# Deep Generative Model based Rate-Distortion for Image Downscaling Assessment

## Abstract

In this paper, we propose a novel measure, namely Image Downscaling Assessment by Rate-Distortion (IDA-RD), to quantitatively evaluate image downscaling algorithms. In contrast to image-based methods that measure the quality of downscaled images, ours is *process-based* that draws ideas from the rate-distortion theory to measure the *distortion* incurred during downscaling. Our main idea is that downscaling and super-resolution (SR) can be viewed as the encoding and decoding processes in the rate-distortion model, respectively, and that a downscaling algorithm that preserves more details in the resulting low-resolution (LR) images should lead to less distorted high-resolution (HR) images in SR. In other words, the distortion should increase as the downscaling algorithm deteriorates. However, it is non-trivial to measure this distortion as it requires the SR algorithm to be *blind* and *stochastic*. Our key insight is that such requirements can be met by recent SR algorithms based on deep generative models that can find all matching HR images for a given LR image on their learned image manifolds. Empirically, we first validate our IDA-RD measure with synthetic downscaling algorithms which simulate distortions by adding various types and levels of degradations to the downscaled images. We then test our measure on traditional downscaling algorithms such as bicubic, bilinear, nearest neighbor interpolation as well as state-of-the-art downscaling algorithms such as DPID (Weber et al., 2016), L0-regularized downscaling (Liu et al., 2017), and Perceptual downscaling (Oeztireli & Gross, 2015). Experimental results show the effectiveness of our IDA-RD in evaluating image downscaling algorithms.

## 1 Introduction

Image downscaling is a fundamental problem in image processing and computer vision. To address the diverse application scenarios, various digital devices with different resolutions, such as smartphones, iPads, and desktop monitors, co-exist, which makes this problem even more important. In contrast to image super-resolution (SR), which aims to "add" information to low-resolution (LR) images, image downscaling algorithms focus on "preserving" information present in the high-resolution (HR) images, which is particularly important for applications and devices with very limited screen spaces.

Traditional image downscaling algorithms low-pass filter an image before resampling it. While this prevents aliasing in the downscaled LR image, important high-frequency details of the HR image are removed simultaneously, resulting in a blurred or overly-smooth LR image. To improve the quality of downscaled images, several sophisticated approaches have been proposed recently, including remapping of high-frequency information (Gastal & Oliveira, 2017), optimization of perceptual image quality metrics (Oeztireli & Gross, 2015), using $L0$-regularized priors (Liu et al., 2017), and pixelizing the HR image (Gerstner et al., 2012; Han et al., 2018; Kuang et al., 2021; Shang & Wong, 2021). Nevertheless, research in image downscaling algorithms has significantly slowed down due to the lack of a quantitative measure to evaluate them. Specifically, standard distance measures (*e.g.* $L1$, $L2$ norm) and full-reference image quality assessment (IQA) methods are not applicable here due to the absence of ground truth LR images; existing No-Reference IQA (NR-IQA) metrics (Mittal et al., 2012b;a; Bosse et al., 2017) cannot be applied either as they rely on the "naturalness" of HR images, which is not present in LR images (we will verify this in our experiments).

In this paper, we propose a new quantitative measure for image downscaling based on Claude Shannon's rate-distortion theory (Berger, 2003), namely Image Downscaling Assessment by Rate-Distortion (IDA-RD). The main idea of our IDA-RD measure is that a superior image downscaling algorithm would try to retain as much information as possible in the LR image, thereby reducing the distortion when being up-scaled (*a.k.a.* super-resolved) to the size of the original HR image. However, such an upscaling method is non-trivial as it must satisfy two challenging requirements: i) *blindness*, *i.e.* it must apply to all kinds of downscaling algorithms without knowing them in advance; ii) *stochasticity*, *i.e.* it must be able to generate a manifold of HR images that captures the conditional distribution of the super-resolution process. Our key insight is that both such requirements can be satisfied by the recent success of deep generative models in blind and stochastic super-resolution. To demonstrate the flexibility of our IDA-RD measure, we show that it can be successfully implemented with two mainstream generative models: Generative Adversarial Networks (Menon et al., 2020) and Normalizing Flows (Lugmayr et al., 2020). Extensive experiments demonstrate the effectiveness of our IDA-RD measure in evaluating image downscaling algorithms. Our contributions include:

- Drawing on Claude Shannon's rate-distortion theory (Berger, 2003), we propose the Image Downscaling Assessment by Rate-Distortion (IDA-RD) measure to quantitatively evaluate image downscaling algorithms, which fills a gap in existing image downscaling research.

- We demonstrate the effectiveness of our IDA-RD measure with extensive experiments on both synthetic and real-world image downscaling algorithms.

## 2 RELATED WORK

**Image Downscaling** has a long history and its traditional methods (*e.g.* bicubic) have now become the standard for image processing and computer vision software, making it difficult to trace their origins. To this end, we only review recent attempts in developing better image downscaling algorithms. For example, Gastal & Oliveira (2017) conducted a discrete Gabor frequency analysis and propose to remap the high-frequency information of HR images to the representable range of the downsampled spectrum, thereby preserving high frequency details in image downscaling. Oeztireli & Gross (2015) model image downscaling as an optimization problem and minimize a perceptual metric (SSIM) between the input and downscaled image. However, the limitations of SSIM are also carried over to their approach. DPID (Weber et al., 2016) preserves small details by assigning higher weights to the input pixels whose color deviates from their local neighborhood within the convolutional filter. Liu et al. (2017) propose an optimization framework using two $L0$ regularized priors that addresses two issues of image downscaling, *i.e.* salient feature preservation and downscaled image construction. Image thumbnailing, a special case of image downscaling, has been studied by Sun & Ling (2013). Their two-component thumbnailing framework, named as Scale and Object Aware Thumbnailing (SOAT) focuses on saliency measure and thumbnail cropping. Li et al. (2018) term image downscaling as image Compact Resolution (CR) and address it with a Convolutional Neural Network (CNN). Inspired by the success of CNNs in image super-resolution (SR), they introduce the CNN-CR model for image downscaling that can be jointly trained with any CNN-SR model. Although their CNN-CR model results in better reconstruction quality than other downscaling algorithms, they only demonstrate results for small downscaling factors ($\times 2$). However, the majority of both image downscaling and super-resolution algorithms tend to focus on larger scaling factors (*e.g.* $\times 8$). Despite the aforementioned works, there does not exist a good quantitative measure for the evaluation of image downscaling methods, which impedes the research on them.

**Image Quality Assessment (IQA)** can be subjective or objective. Between them, subjective methods rely on the visual inspection by human assessors while objective methods resort to quantitative measures, *e.g.*, image statistics. Examples of the most commonly used objective IQA metrics include Peak Signal-to-Noise Ratio (PSNR), Structural Similarity Index Measure (SSIM), Multi-Scale SSIM (MS-SSIM) (Wang et al., 2003) and Learned Perceptual Image Patch Similarity (LPIPS) (Zhang et al., 2018). However, such IQA metrics are not applicable in the evaluation of image downscaling algorithms as there are no ground truth LR images for comparison. Thus, most researchers rely on subjective evaluation of downscaled images, which is costly and time-consuming.

**No-Reference Image Quality Assessment** (NR-IQA) addresses IQA in the absence of a reference (*i.e.* ground truth) image. For example, Mittal et al. (2012a) propose BRISQUE, an NR-IQA metric that uses the natural scene statistics (NSS) to quantify loss of "naturalness" in distorted images.

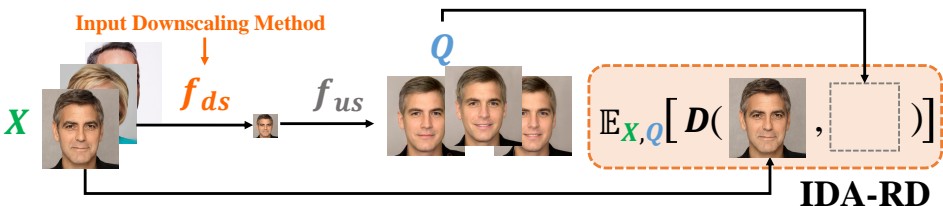

Figure 1: Illustration of the proposed IDA-RD measure. Given a downscaling method $f_{ds}$ to be evaluated, i) we first use it to downscale several HR images; ii) then, we upscale them back to the original resolution with $f_{us}$ and measure the distortion from the corresponding HR images. Such an upscaling method leverages the recent success in deep generative models and thus can i) apply to arbitrarily down-scaled images and ii) output a manifold of HR images that captures the conditional distribution given a downscaled image.

Using locally normalized luminances, BRISQUE models a regressor which maps the feature space to image quality scores. Based on their NSS, Mittal et al. (2012b) further devised an Opinion Unaware (OU) and Distortion Unaware (DU) model for blind NR-IQA, which is named as NIQE. Bosse et al. (2017) follow a data-driven approach for NR-IQA. Inspired by Siamese networks, they train a deep neural network for feature extraction and regression in an end-to-end manner. However, due to the lack of a large enough training dataset, their model does not generalize well. However, such NQ-IQA metrics are also not applicable, as the "naturalness" they rely on exists only in HR but not LR images. To this end, we borrow ideas from Claude Shannon's rate-distortion theory and propose a new measure called Image Downscaling Assessment by Rate-Distortion (IDA-RD). Our IDA-RD measure leverages the recent success in deep generative models and shows promising results in the quantitative evaluation of image downscaling methods.

**Deep Generative Models** We refer interested readers to Bond-Taylor et al. (2021) for a detailed survey on deep generative modeling. Here, we review the two deep generative models used in our work, *i.e.* Generative Adversarial Networks (GANs) and normalizing flows. Since the pioneering work by Goodfellow et al. (2014), GANs have experienced significant improvements. For example, Radford et al. (2015) proposed DCGAN, which incorporates convolutional neural networks for better image synthesis. Arjovsky et al. (2017) addressed the notorious instability of GAN training by employing a novel loss function, *i.e.* the Wasserstein distance loss. To date, the StyleGAN series (Karras et al., 2019; 2020; 2021) developed by Nvidia has shown impressive (maybe even the best) results in high-resolution and high-quality image synthesis, leading to various applications in image processing and manipulation (Abdal et al., 2019; 2020; Zhu et al., 2020). In this paper, we follow Menon et al. (2020) and implement our measure with a StyleGAN generator pre-trained on portrait images. Nevertheless, normalizing flows (Rezende & Mohamed, 2015; Papamakarios et al., 2021; Keller et al., 2021) that construct complex distributions by transforming a probability density function through a series of invertible mappings have attracted increasing attention in the past several years. In this paper, we employ the SRFlow (Lugmayr et al., 2020) model to implement our measure, which directly learns the conditional distribution of the HR output given the LR input.

## 3 OUR APPROACH

In this section, we first introduce the definition of our metric derived from Claude Shannon's rate-distortion theory (Berger, 2003), and then detail how *deep generative models* help to sidestep the data scarcity challenge that impedes the application of the proposed metric.

### 3.1 METRIC DEFINITION

We create a proxy task, namely the *lossy compression problem* underpinned by Claude Shannon's rate-distortion theory (Berger, 2003), and formulate image downscaling as its encoding process:

$$\inf_{Q_f(\hat{x}|x)} \mathbb{E}[D_Q(X, \hat{X})] \ s.t. \ I_Q(X; \hat{X}) \leq R \tag{1}$$

where $X$ is the set of input high-resolution images, $\hat{X}$ is the set of output reconstructed images, $R$ is a rate constraint determined by the downscaling process[1], $Q_f(\hat{x}|x)$ or $Q$ for short is the probability density function (PDF) of reconstructed HR images $\hat{x}$ conditioned on an input HR image $x$ with respect to a given lossy image reconstruction function $f$ that $\hat{x} = f(x) = f_{us}(f_{ds}(x))$, where $f_{us}$ and $f_{ds}$ denote image upscaling and downscaling functions respectively, $D_Q$ is a distortion metric between two image sets where the image correspondence is determined by $Q$. Thus, we propose to use the expectation of the distortion as an evaluation metric for image downscaling:

$$S(f_{ds}) = \mathbb{E}[D_Q(X, \hat{X})] = \mathbb{E}_x\{\mathbb{E}_{\hat{x}|x}[D(x, \hat{x})]\}, \tag{2}$$

where $x \in X, \hat{x} \in \hat{X}$, $D$ is a distortion metric between two images, *e.g.*, LPIPS (Zhang et al., 2018). The lower $S$, the better the downscaling algorithm $f_{ds}$. Although straightforward, the application of such a metric remained a challenge in the past as it requires a strong upscaling function $f_{us}$ that can:

- Reconstruct the input image $x$ regardless of the input downscaling algorithm $f_{ds}$.
- Generate a conditional distribution of reconstructed images $\hat{x}|x$ for each $x$.

Between them, the first is commonly known as *blind image super-resolution* that is essentially a many-to-one mapping problem that aims to map different distorted downscaled images to the same high-resolution image; the second is commonly known as *one-to-many super-resolution* due to its ill-posed nature caused by the information loss during downscaling (Lugmayr et al., 2020).

**Data Scarcity Challenge** Combining the above two requirements makes the desired $f_{us}$ an extremely challenging many-to-many mapping problem that has remained unsolved for decades. Specifically, the numerous kinds of distorted downscaled images and the corresponding countless high-resolution images for each of them makes it infeasible to collect sufficient data for supervised learning methods:

$$f_{us} = \arg\min_{f_\theta} \mathbb{E}_{I_{LR}}(\mathbb{E}_{I_{HR}}||I_{HR} - f_\theta(I_{LR})||) \tag{3}$$

where $I_{HR}$ and $I_{LR}$ denote the high-resolution (HR) and low-resolution (LR) training images respectively, $\mathbb{E}_{I_{HR}}$ indicates that there are many $I_{HR}$ corresponding to the same $I_{LR}$, $\mathbb{E}_{I_{LR}}$ indicates that there are many $I_{LR}$ obtained by different image downscaling methods $f_{ds}$.

## 3.2 EVALUATION WITH DEEP GENERATIVE MODELS

Our key insight is that the above-mentioned data scarcity challenge (Eq. 3) can be overcome by the recent successes in deep generative modeling (Goodfellow et al., 2014; Radford et al., 2015; Arjovsky et al., 2017; Karras et al., 2019; 2020; 2021; Rezende & Mohamed, 2015; Papamakarios et al., 2021; Keller et al., 2021). In deep generative modeling, a neural network model is trained to learn a manifold of natural and high-resolution (HR) images from samples in the training dataset. This has been successfully applied to various image processing tasks (Abdal et al., 2019; 2020; Zhu et al., 2020). To demonstrate the flexibility of our metric, we show its two implementations using two mainstream deep generative models: i) Generative Adversarial Networks (GANs) and ii) Normalizing Flows respectively as follows.

**Implementation with a GAN generator.** Similar to Menon et al. (2020), we implement the upsampling function $f_{us}$ in our metric using an optimization-based GAN inversion method (Abdal et al., 2019; 2020). Leveraging the power of a pre-trained StyleGAN (Karras et al., 2019) generator $G$, we define our GAN-based $f_{us}$ (Eq. 2) as locating the optimized StyleGAN latent code $\mathbf{z_i^*}$ so that its corresponding HR image $G(\mathbf{z_i^*})$ synthesized by $G$ shares the same downscaled image as an input LR image $I_{LR} = f_{ds}(x)$:

$$f_{us}(I_{LR}, i) = G(\mathbf{z_i^*}) = \arg\min_{G(\mathbf{z_i})} ||I_{LR} - f_{ds}(G(\mathbf{z_i}))|| \tag{4}$$

where $I_{LR} = f_{ds}(x)$ denotes the input LR image downscaled by $f_{ds}$, $\mathbf{z_i}$ denotes the $i$-th randomly initialized latent code to be optimized to get the $i$-th sample from $\hat{x}|I_{LR}$ (*i.e.*, $G(\mathbf{z_i^*})$), $i = 1, 2, 3, ...$ is the index. It can be observed that i) our $f_{us}$ sidesteps the data scarcity challenge (Eq. 3) by using

---

[1]Note that in image downscaling, such a constraint on $R$ is always satisfied as the downscaled images are of a fixed resolution defined by users.

a StyleGAN generator that is trained with HR images only (*i.e.*, without any many-to-many LR-HR training pairs); ii) it relocates the supervision to downscaling (*i.e.*, enforcing different HR images to be downscaled to the same LR image) and thus outputs high quality HR images $G(\mathbf{z_i^*})$ that applies to an arbitrary choice of $f_{ds}$; iii) it is inherently stochastic given the random choices of $\mathbf{z_i}$.

**Implementation with a Flow model.** We employ the SRFlow model (Lugmayr et al., 2020) and implement the $f_{us}$ in our metric with a conditional invertible neural network. Leveraging its invertible nature, $f_{us}$ is trained to explicitly learn the conditional distribution $\hat{x}|I_{LR}$ by minimizing the negative log-likelihood:

$$f_{us} = \arg\min_{f_\theta} -\log p_{\mathbf{z}}(f_\theta(x|I_{LR})) \tag{5}$$

where $I_{LR} = f_{ds}^{\text{bicubic}}(x)$ is a bicubic downscaled image of HR input $x$, $\mathbf{z}$ denotes a random latent variable whose distribution encodes $\hat{x}|I_{LR}$ with a 'reparameterization trick'. Although trained with only bicubic downscaling, surprisingly, we observed that the resulting $f_{us}$ can also be applied to evaluate other downscaling methods.

We use SRFlow in the final version of our metric as it shares similar performance as the GAN-based implementation but has a much lower time cost. Please see Sec. 4.4 for a detailed ablation study.

## 4 EXPERIMENTS

To validate the effectiveness of our IDA-RD measure, we first test it with synthetic image downscaling methods whose performance are known beforehand (Sec. 4.2). Specifically, we simulate different types and levels of downscaling distortions by adding controllable degradations (*e.g.*, Gaussian Blur, Contrast Change) to bicubic-downscaled images. In principle, the heavier the degradation, the worse the results of downscaling, and the higher our measure should be. We also validate the effectiveness of our IDA-RD measure across different scaling factors. Then, we show that our measure can also be used to evaluate real-world image downscaling methods like Bicubic, Bilinear, Nearest Neighbour, and state-of-the-art downscaling methods like L0-regularized (Liu et al., 2017), Perceptual (Oeztireli & Gross, 2015) and DPID (Weber et al., 2016) (Sec. 4.3). Third, we perform a thorough ablation study to justify the algorithmic choices of our measure (Sec. 4.4). Finally, we empirically justify our motivation in Sec. 4.5. Please see the appendix for additional experiments and examples of downscaled images (Appendix A.1).

### 4.1 EXPERIMENTAL SETUP

**Dataset** Unless specified, we use a balanced subset of 900 images from the FFHQ dataset (Karras et al., 2019), including face images at $1024\times1024$ resolution, as the set of input high-resolution images $X$ in Eq. 2 for our IDA-RD measure. Please see Appendix A.2 for more details on how we construct balanced subsets of images from FFHQ. The results on other datasets, including NPRportrait 1.0 (Rosin et al., 2022) and AFHQ-Cat (Choi et al., 2020), are shown in Sec. 4.4. Note that we use these domain-specific datasets as they are more stable for SRFlow. Please see Appendix A.8 for the results and discussions on real-world datasets, *e.g.*, DIV2K (Agustsson & Timofte, 2017), Flickr30k (Young et al., 2014).

**Image Upscaling Algorithms** We use SRFlow (Lugmayr et al., 2020) as the $f_{us}$ in Eq. 2. Specifically, we used the models provided by the authors for $4\times$ and $8\times$ super resolution that are pre-trained on DIV2K (Agustsson & Timofte, 2017) and Flickr2K datasets[2]. Unless specified, we use the $8\times$ model for all experiments. Note that we also tested PULSE (Menon et al., 2020) as an alternative in Sec. 4.4. For PULSE, we use the same StyleGAN generator pre-trained with FFHQ (Karras et al., 2019). This model generates face images of size $1024\times1024$. We use a learning rate of $0.4$, and stop the optimization for each image after 200 steps of spherical gradient descent. The noise signals of the StyleGAN generator were kept fixed.

**Hyperparameters** Unless specified, we use $N_Q = 5$ as the number of images upscaled from a single downscaled image for the estimation of $Q$ in Eq. 2; we use LPIPS (Zhang et al., 2018) as the distortion measure $D$ in Eq. 2; we use $N_X = 900$ as the number of images in the set of high-resolution image $X$ in Eq. 2.

---

[2]https://github.com/andreas128/SRFlow

Table 1: IDA-RD scores for synthetic image downscaling with different types and levels of degradations (a), (b); with mixed degradations (c). The numbers in parentheses denote degradation parameters. As a reference, the IDA-RD score for the bicubic-downscaled image without degradation is $0.11\pm0.145$. It is best to **Zoom In** to view the examples of downscaled images with different types and levels of degradations.

| | Gauss. Blur | | Gauss. Noise | | Contrast Inc. | | Contrast Dec. |
|---|---|---|---|---|---|---|---|
| (1.0) | $0.320\pm0.048$ | (0.05) | $0.482\pm0.051$ | (1.5) | $0.231\pm0.042$ | (0.75) | $0.330\pm0.047$ |
| (2.0) | $0.434\pm0.057$ | (0.10) | $0.640\pm0.052$ | (2.0) | $0.317\pm0.041$ | (0.50) | $0.644\pm0.074$ |
| (4.0) | $0.579\pm0.065$ | (0.20) | $0.659\pm0.052$ | (2.5) | $0.462\pm0.043$ | (0.25) | $0.669\pm0.034$ |

(a)

| | Quantization |
|---|---|
| (15) | $0.164\pm0.002$ |
| (10) | $0.205\pm0.003$ |
| (5) | $0.463\pm0.064$ |

(b)

| Gauss. Blur (1) | $0.320\pm0.048$ |
|---|---|
| + Gauss. Noise (0.05) | $0.585\pm0.062$ |
| + Contrast Dec. (0.75) | $0.664\pm0.046$ |
| + Quantization (10) | $0.795\pm0.063$ |

(c)

Table 2: IDA-RD scores for synthetic image downscaling methods with different scaling factors. $(\cdot)$: the resolution of downscaled images. Bicubic: bicubic-downscaled image without degradation. G.B.: Gaussian Blur. The $32\times$ super-resolution is achieved by a concatenation of a $8\times$ and a $4\times$ upscaling implemented by pretrained SRFlow models.

| Scaling Factor | Bicubic | G.B. ($\sigma = 1.0$) | G.B. ($\sigma = 2.0$) | G.B. ($\sigma = 4.0$) |
|---|---|---|---|---|
| $4\times$ ($256 \times 256$) | $0.058\pm0.142$ | $0.146\pm0.032$ | $0.269\pm0.043$ | $0.412\pm0.055$ |
| $8\times$ ($128 \times 128$) | $0.110\pm0.145$ | $0.320\pm0.048$ | $0.434\pm0.057$ | $0.579\pm0.065$ |
| $32\times$ ($32 \times 32$) | $0.228\pm0.056$ | $0.614\pm0.068$ | $0.680\pm0.066$ | $0.741\pm0.065$ |

## 4.2 TEST WITH SYNTHETIC DOWNSCALING METHODS

In this section, we demonstrate the effectiveness of our IDA-RD measure by testing its performance on synthetic downscaling methods, which simulate the effects of different downscaling methods by adding controllable degradations to bicubic-downscaled images.

### 4.2.1 EFFECTIVENESS ACROSS DEGRADATION TYPES AND LEVELS

As detailed below, we test our IDA-RD measure with four sets of synthetic downscaling methods that apply different types and levels of degradations to bicubic-downscaled images respectively.

**Gaussian Blur.** We apply Gaussian blur to the bicubic-downscaled images. The standard deviation of the blur kernel $\sigma$ is chosen from $\{1.0, 2.0, 4.0\}$. The kernel size was set as $(3\sigma + 1)$. The results are shown in Table 1 (a).

**Gaussian Noise.** We add Gaussian noise to the bicubic-downscaled images. The standard deviation $\sigma$ of the noise is chosen from $\{0.05, 0.1, 0.2\}$. The results are shown in Table 1 (a).

**Contrast Change.** We apply contrast change to bicubic-downscaled images. To increase the contrast, we select the scale factor from $\{1.5, 2.0, 2.5\}$. Note that such scaling can cause degradation due to the clipping of extreme intensity values. Similarly, to decrease the contrast, we select the contrast parameter from $\{0.25, 0.50, 0.75\}$. The results are shown in Table 1 (a).

**Quantization.** We apply pixel quantization to bicubic-downscaled images and select the number of color thresholds from $\{5, 10, 15\}$. Specifically, we apply Otsu's multilevel thresholding algorithm (Otsu, 1979) to the graylevel histogram which is derived from the color image, and then apply these thresholds uniformly to each of the RGB color channels. The results are shown in Table 1 (b).

**Mixed Degradations.** In addition to single degradations mentioned above, we also demonstrate the effectiveness of our IDA-RD measure on their mixtures. The results are shown in Table 1 (c).

Table 3: IDA-RD scores for real-world image downscaling methods with different scaling factors. S.F.: Scaling Factor, the resolutions of downscaled images (*e.g.*, 512×512 for 2×, 64×64 for 16×), are omitted for simplicity. N.N.: Nearest Neighbour. $L0$-reg.: L0-regularized. Note that the relatively large standard deviations in some cases (especially when the scaling factors are small) indicate the algorithmic biases of image downscaling methods against individual images, *e.g.*, flat images with large color blocks may suffer less from information loss. The 32× super-resolution is achieved by a concatenation of a 8× and a 4× upscaling implemented by pretrained SRFlow models.

| S.F. | Bicubic | Bilinear | N.N. | DPID | Perceptual | $L0$-reg. |
|------|---------|----------|------|------|------------|-----------|
| 4× | 0.058±0.142 | 0.031±0.053 | 0.335±0.310 | 0.122±0.234 | 0.388±0.321 | 0.136±0.251 |
| 8× | 0.110±0.145 | 0.090±0.067 | 0.512±0.340 | 0.127±0.294 | 0.398±0.337 | 0.213±0.301 |
| 32× | 0.228±0.056 | 0.272±0.056 | 0.601±0.163 | 0.291±0.076 | 0.514±0.152 | 0.307±0.050 |

Table 4: Ablation study of $f_{us}$, the image upscaling algorithms. PULSE (Menon et al., 2020) and SRFlow (Lugmayr et al., 2020) have similar results but those of SRFlow are more distinguishable.

| | Bicubic | Bilinear | N.N. | DPID | Perceptual | $L0$-reg. |
|------|---------|----------|------|------|------------|-----------|
| PULSE | 0.171±0.015 | 0.164±0.015 | 0.254±0.018 | 0.179±0.016 | 0.223±0.017 | 0.205±0.016 |
| SRFlow | 0.110±0.145 | 0.090±0.067 | 0.512±0.340 | 0.127±0.294 | 0.398±0.337 | 0.213±0.301 |

It can be observed that our IDA-RD measure works as expected (*i.e.*, the stronger the degradation, the worse the downscaling algorithm, and the higher the IDA-RD) for all synthetic image downscaling methods, which demonstrates its effectiveness.

### 4.2.2 EFFECTIVENESS ACROSS SCALE FACTORS

We further demonstrate the effectiveness of our IDA-RD measure on synthetic downscaling algorithms across different scaling factors. As Table 2 shows, we test our IDA-RD on synthetic downscaling algorithms of different levels of Gaussian Blur degradation as mentioned above. It can be observed that: i) the larger the scaling factor, the more the information loss, and the higher the IDA-RD; ii) the stronger the degradation, the worse the downscaling algorithm, and the higher the IDA-RD; which justifies the validity of our IDA-RD measure.

### 4.3 EVALUATING EXISTING DOWNSCALING METHODS

We apply our method to compare six existing downscaling algorithms, consisting of three traditional methods: Bicubic, Bilinear, Nearest Neighbor (N.N.), and three state of the art methods: DPID (Weber et al., 2016), L0-regularized downscaling (Liu et al., 2017), and Perceptual (Oeztireli & Gross, 2015) downscaling. The results are shown in Table 3. It can be observed that: i) when applied to classical downscaling algorithms (*i.e.*, Bicubic, Bilinear, and N.N.), our IDA-RD measure identifies the quality of these algorithms in the correct order (Bilinear > Bicubic > N.N.), although the difference between the results of Bicubic and Bilinear downscaling is not significant as expected; ii) when applied to SOTA ones, the common belief is that these algorithms should perform better than Bilinear downscaling. However, none of these methods achieve a better in IDA-RD, suggesting that although SOTA image downscaling methods excel in perceptual quality, they actually lose more information than Bilinear downscaling. Nevertheless, it can be observed that DPID and L0-regularized methods are slightly better than Perceptual downscaling on our IDA-RD measure, which is consistent with previous understanding. These indicate that our IDA-RD measure is a useful complement to visual inspection, *i.e.*, a good image downscaling algorithm should be both visually satisfying and achieve a low IDA-RD score, which further validates the role of our measure in providing new insights into image downscaling algorithms. Please see Appendix A.10 for a qualitative comparison.

### 4.4 ABLATION STUDY

In this experiment, we justify the algorithmic choices of our IDA-RD measure, *i.e.*, $f_{us}$, $D$, the number of images used to estimate $Q$ and in $X$, and the content of $X$ in Eq. 2, by performing a thorough ablation study on them.

Table 5: Ablation study of $N_Q$, the number of images required for a robust estimation of $Q$ in Eq. 2.

| $N_Q$ | 1 | 3 | 5 | 10 | 15 |
|---|---|---|---|---|---|
| Bicubic | 0.103±0.141 | 0.109±0.142 | 0.110 ±0.145 | 0.111±0.145 | 0.110±0.145 |
| Bilinear | 0.090±0.069 | 0.090±0.067 | 0.090 ±0.067 | 0.090±0.067 | 0.090±0.067 |
| N.N. | 0.513±0.341 | 0.512±0.340 | 0.512 ±0.340 | 0.512±0.340 | 0.511±0.340 |

Table 6: Ablation study of $D$, the distortion measure in Eq. 2. Dec.: Decrease. Param.: Parameter. Please see Table 14 in Appendix A.3 for experiments with other synthetic downscaling methods.

| | Param. | PSNR | SSIM | MS-SSIM | LPIPS |
|---|---|---|---|---|---|
| | 0.75 | 22.137±4.020 | 0.834±0.159 | 0.881±0.101 | 0.330±0.047 |
| Contrast Dec. | 0.50 | 17.814±2.148 | 0.714±0.087 | 0.819±0.080 | 0.644±0.074 |
| | 0.25 | 14.790±1.461 | 0.578±0.072 | 0.579±0.028 | 0.669±0.034 |
| | 1.00 | 25.159±1.999 | 0.744±0.059 | 0.929±0.017 | 0.320±0.048 |
| Gaussian Blur | 2.00 | 22.365±1.875 | 0.646±0.073 | 0.849±0.033 | 0.434±0.057 |
| | 4.00 | 19.738±1.739 | 0.558±0.080 | 0.715±0.051 | 0.579±0.065 |

**Choice of $f_{us}$.** As Table 4 shows, both PULSE (Menon et al., 2020) and SRFlow (Lugmayr et al., 2020) have similar results when used as $f_{us}$ in our IDA-RD measure, *i.e.*, N.N. > Perceptual > L0-regularized > DPID > Bicubic > Bilinear. However, since SRFlow yields more distinguishable results and runs much faster (Table 15 in Appendix A.4), we use it in our IDA-RD measure. Nevertheless, our IDA-RD is very flexible (*i.e.*, not restricted to PULSE or SRFlow) and will benefit from future progresses of blind and stochastic super-resolution methods (please see Appendix A.9).

**Number of Images used to Estimate $Q$.** As Table 5 shows, for a downscaled image, we investigate how many images are required to be upscaled from it (by $f_{us}$) to achieve a robust estimation of the conditional distribution $Q$ and thus our IDA-RD, namely $N_Q$. It can be observed that the results become stable when $N_Q \geq 5$, so we choose $N_Q = 5$ for our IDA-RD measure.

**Choice of $D$.** As Table 6 shows, we test different choices of $D$ including multiple image distortion metrics: Peak Signal-to-Noise Ratio (PSNR), Structural Similarity Index Measure (SSIM) (Wang et al., 2004), MS-SSIM (Multi-Scale SSIM), and LPIPS (Zhang et al., 2018). Experimental results demonstrate a similar trend across all of them, indicating the flexibility of our IDA-RD measure. Nevertheless, since LPIPS is a more advanced metric that has been shown to be more consistent with human perception, we use it in the final version of our IDA-RD measure.

**Number of Images in $X$.** As Table 7 shows, we investigate how many images are required in the test dataset $X$ consisting of high-resolution images to achieve a robust estimation of IDA-RD, namely $N_X$. It can be observed that the results become stable when $N_X \geq 900$, so we choose $N_X = 900$ for our IDA-RD measure.

Table 7: Ablation study of $N_X$, the number of images in test dataset $X$ in Eq. 2. Synthetic image downscaling methods with Contrast Decrease with $\sigma = 0.75$ (DG1); Gaussian Noise with $\sigma = 0.05$ (DG2); mixed noise consisting of Gaussian Blur with $\sigma = 1.0$, Contrast Decrease with $\sigma = 0.75$, and Gaussian Noise with $\sigma = 0.05$ (DG3); are used in the experiments.

| $N_X$ | 30 | 300 | 600 | 900 | 1200 | 1500 |
|---|---|---|---|---|---|---|
| DG1 | 0.320±0.026 | 0.321±0.047 | 0.321±0.046 | 0.330±0.047 | 0.325±0.047 | 0.329±0.047 |
| DG2 | 0.501±0.055 | 0.473±0.051 | 0.481±0.050 | 0.482±0.051 | 0.483±0.051 | 0.484±0.051 |
| DG3 | 0.483±0.088 | 0.312±0.048 | 0.321±0.045 | 0.320±0.048 | 0.321±0.047 | 0.322±0.048 |

**The Content of $X$.** As Table 8 shows, in addition to FFHQ (Karras et al., 2019), we test our IDA-RD measure on another two datasets: the NPRportrait 1.0 benchmark set (Rosin et al., 2022) and AFHQ-Cat (Choi et al., 2020). Between them, we use all 60 images at around 800×1024 resolution from the NPRportrait 1.0 benchmark set as $X$, which was carefully constructed so as to include a controlled diversity of gender, age and ethnicity; we use a random sample of 900 images at 512×512 resolution from the AFHQ-Cat dataset as $X$. We test them with 4× image downscaling. It can be

Table 8: Ablation study of the contents of dataset $X$ in Eq. 2. (1) Bicubic (2) Bilinear (3) Nearest Neighbor (N.N.) (4) DPID (5) Perceptual (6) $L0$-regularized.

|  | FFHQ | NPRportrait 1.0 | AFHQ-Cat |
|---|---|---|---|
| (1) | 0.110±0.145 | 0.119±0.166 | 0.107±0.029 |
| (2) | 0.090±0.067 | 0.100±0.101 | 0.091±0.033 |
| (3) | 0.512±0.340 | 0.329±0.292 | 0.277±0.103 |
| (4) | 0.127±0.294 | 0.119±0.099 | 0.152±0.047 |
| (5) | 0.398±0.337 | 0.391±0.231 | 0.289±0.067 |
| (6) | 0.213±0.301 | 0.166±0.234 | 0.211±0.025 |

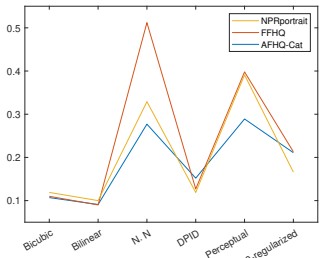

observed that our conclusions hold for all datasets, which further verifies the flexibility of our method against the content of $X$. Without loss of generality, we use FFHQ in our IDA-RD measure.

### 4.5 MOTIVATION JUSTIFICATION

**Invalidity of NR-IQA Metrics** As Table. 9 shows, existing NR-IQA metrics, such as NIQE (Mittal et al., 2012b) and BRISQUE (Mittal et al., 2012a), are not suitable for the image downscaling problem, especially extreme downscaling. It can be observed that i) NIQE struggles to calculate proper scores at all resolutions below 128×128; ii) BRISQUE does not provide the correct scores at a resolution of 32×32. Please see Appendix A.5 for results on higher resolutions.

Table 9: NIQE and BRISQUE scores at different resolutions. The test image was randomly selected from the FFHQ dataset and bicubic-downscaled to different resolutions (LR). Different levels of Gaussian Blur with kernel $\sigma = 1.0, 2.0, 4.0$ were applied as synthetic image downscaling methods.

|  | Resolution | LR | $\sigma = 1.0$ | $\sigma = 2.0$ | $\sigma = 4.0$ |
|---|---|---|---|---|---|
| | 128×128 | 18.873 | 18.872 | 18.870 | 18.869 |
| NIQE↓ | 64×64 | 18.872 | 18.872 | 18.870 | 18.869 |
| | 32×32 | 18.873 | 18.869 | 18.870 | 18.867 |
| | 128×128 | 16.045 | 34.423 | 47.017 | 55.166 |
| BRISQUE↓ | 64×64 | 41.360 | 42.417 | 43.346 | 54.344 |
| | 32×32 | 43.458 | 43.458 | 44.015 | 43.668 |

**Invalidity of Non-blind and Non-stochastic SR method** As Table 10 shows, non-blind and non-stochastic SR methods like ESRGAN (Wang et al., 2018) and SR3 (Saharia et al., 2022) fail to distinguish among image downscaling algorithms, which justifies the choice of blind and stochastic SR methods in our IDA-RD.

|  | Bicubic | Bilinear | N.N. | DPID | Perceptual | $L0$-reg. |
|---|---|---|---|---|---|---|
| ESRGAN | 0.022±0.012 | 0.017±0.006 | 0.058±0.016 | 0.025±0.009 | 0.024±0.004 | 0.024±0.007 |
| SR3 | 0.169±0.048 | 0.164±0.047 | 0.179±0.040 | 0.171±0.044 | 0.172±0.043 | 0.171±0.049 |

Table 10: Invalidity of using ESRGAN and SR3 in our IDA-RD measure.

### 5 CONCLUSION

In this paper, we presented Image Downscaling Assessment by Rate Distortion (IDA-RD), a quantitative measure for the evaluation of image downscaling algorithms. Our measure circumvents the requirement of a ground-truth LR image by measuring the distortion in the HR space, which is enabled by the recent success of blind and stochastic super-resolution algorithms based on deep generative models. We validate our approach by testing various synthetic downscaling algorithms, simulated by adding degradations, on various datasets. We also test our measure on real-world image downscaling algorithms, which further validates the role of our measure in providing new insights into image downscaling algorithms. Please see Appendix A.6 for Limitation and Future Work.

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

## A APPENDIX

### A.1 EXAMPLES OF DOWNSCALED IMAGES USED IN OUR EXPERIMENTS

Table 11 and Table 12 show examples of images downscaled by synthetic and real-world image downscaling methods used in our experiments, respectively.

### A.2 BALANCING FFHQ INTO AGE-, GENDER-, AND RACE-BALANCED SUBSETS

We balance the FFHQ dataset Karras et al. (2019) into subsets (*i.e.*, $X$ in Eq. 2) that are balanced in age, gender and ethnicity for a fair evaluation of our IDA-RD measure. For the gender and age labels of FFHQ images, we use those offered by the FFHQ-features-dataset[3]; for the ethnicity labels of FFHQ images, we use the recognition results of DeepFace[4]. According to the above, we define i) four age groups: Minors (0-18), Youth (19-36), Middle Aged (36-54) and Seniors (54+); ii) three major ethnic groups: Asian, White and Black; iii) two gender groups: Male and Female. We apply K-means to cluster FFHQ images in 24 (4×3×2) groups and select images from them evenly to generate the subsets used in our experiments. As Table 13 shows, the subsets used in our experiments are highly-balanced in terms of age, gender and ethnicity.

### A.3 ADDITIONAL ABLATION STUDY ON $D$ THE DISTORTION MEASURE

As a complement to Table 6 in the main paper, Table 14 shows additional results for the ablation study of $D$, which further justifies our choice of LPIPS as the distortion measure in our IDA-RD.

---

[3]https://github.com/DCGM/ffhq-features-dataset
[4]https://github.com/serengil/deepface

Table 11: Examples of images downscaled by synthetic image downscaling methods, *i.e.*, those adds controllable degradations to bicubic-downscaled images (Sec. 4.2). The numbers below images are the degradation parameters. LR: bicubic-downscaled images, Dec.: decrease, Inc.: increase, Gauss.: Gaussian.

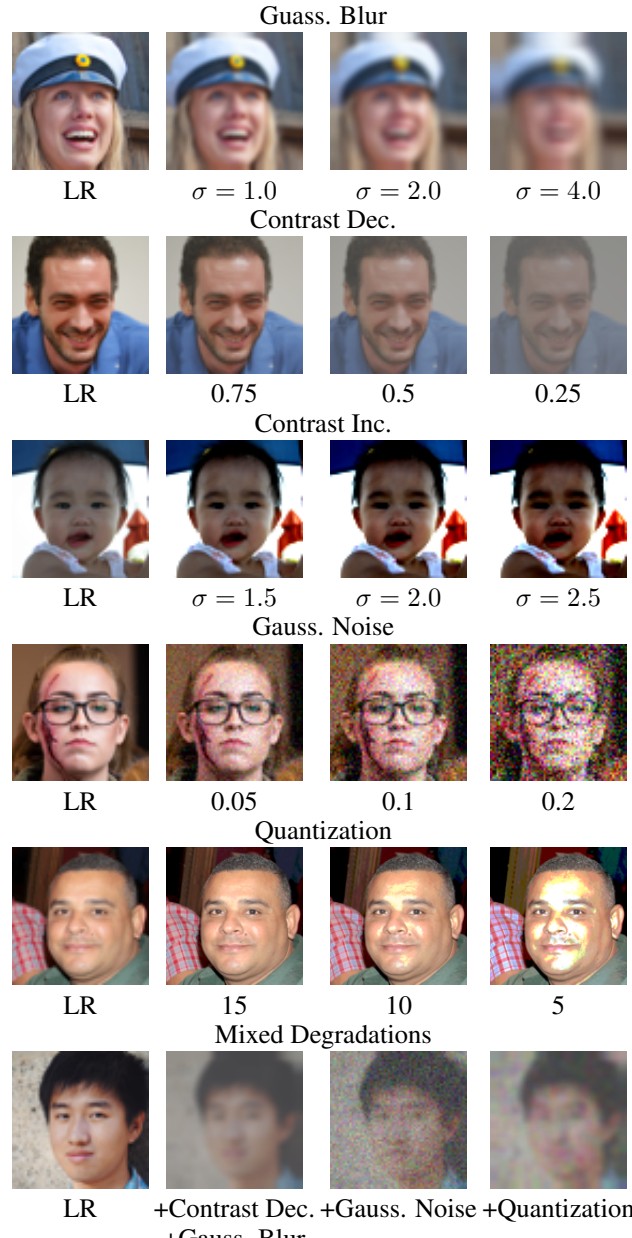

## A.4 TIME COMPLEXITY

Table 15 shows the running times of our IDA-RD measure using PULSE and SRFlow as $f_{us}$ (Eq. 2) on an Nvidia RTX3090 GPU, respectively. It can be observed that the SRFlow implementation runs much faster, which justifies our choice of using it in our IDA-RD measure.

Table 12: Examples of images downscaled by real-world image downscaling methods. N.N.: Nearest Neighbour; $L0$-reg.: L0-regularized.

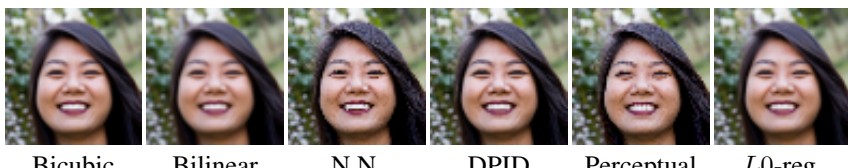

| Bicubic | Bilinear | N.N. | DPID | Perceptual | $L0$-reg. |

Table 13: Statistics of our balanced FFHQ subsets. MI: Minors, Y: Youth, MA: Middle Aged, S: Senior; A: Asian, W: White, B: Black; M: Male, F: Female. J.E.: Joint Entropy, which measures the extent to which a subset is balanced. As a reference, a fully-balanced subset has a joint entropy of $-24 * (1/24) * \log_2(1/24) \approx 4.5850$.

| Size | Age | | | | Ethnicity | | | Gender | | J.E. |
|---|---|---|---|---|---|---|---|---|---|---|
| | MI | Y | MA | S | A | W | B | M | F | |
| 30 | 6 | 9 | 7 | 8 | 10 | 10 | 10 | 15 | 15 | 4.2817 |
| 300 | 76 | 75 | 70 | 79 | 102 | 100 | 98 | 150 | 150 | 4.4998 |
| 600 | 168 | 142 | 141 | 149 | 200 | 194 | 206 | 329 | 271 | 4.5245 |
| 900 | 222 | 227 | 215 | 236 | 304 | 295 | 301 | 452 | 448 | 4.5343 |
| 1200 | 445 | 442 | 453 | 460 | 608 | 591 | 601 | 902 | 898 | 4.5375 |
| 1500 | 684 | 664 | 673 | 679 | 909 | 887 | 904 | 1352 | 1348 | 4.5386 |

## A.5 ADDITIONAL RESULTS WITH NIQE AND BRISQUE

As a complement to Table 9 in the main paper, Table 16a and Table 16b show additional results of NIQE (Mittal et al., 2012b) and BRISQUE (Mittal et al., 2012a) at higher resolutions where the two scores work better.

## A.6 LIMITATION AND FUTURE WORK

**Limitations.** Since our measure makes use of GAN- and Flow-based super-resolution (SR) models, the limitations of these models are carried over as well. First of all, we cannot use test data beyond the learnt distribution of the SR model. For example, unlike the SRFlow (Lugmayr et al., 2020) model trained on general images that are used in the main paper, our GAN-based implementation uses a StyleGAN generator pre-trained on portrait images, which only allows for the use of portrait face images to evaluate downscaling algorithms. Also, although highly unlikely to occur, we cannot

Table 14: Ablation study of $D$, the distortion measure in Eq. 2. Dec.: Decrease. Param.: Parameter.

| | Param. | PSNR | SSIM | MS-SSIM | LPIPS |
|---|---|---|---|---|---|
| Contrast Dec. | 0.75 | 22.137±4.020 | 0.834±0.159 | 0.881±0.101 | 0.330±0.047 |
| | 0.50 | 17.814±2.148 | 0.714±0.087 | 0.819±0.080 | 0.644±0.074 |
| | 0.25 | 14.790±1.461 | 0.578±0.072 | 0.579±0.028 | 0.669±0.034 |
| Contrast Inc. | 1.50 | 16.641±4.019 | 0.603±0.223 | 0.772±0.150 | 0.231±0.042 |
| | 2.00 | 13.450±3.539 | 0.482±0.192 | 0.693±0.131 | 0.317±0.041 |
| | 2.50 | 11.032±2.893 | 0.357±0.159 | 0.602±0.120 | 0.462±0.043 |
| Gaussian Noise | 0.05 | 20.784±0.160 | 0.597±0.004 | 0.648±0.071 | 0.482±0.051 |
| | 0.10 | 18.121±1.713 | 0.563±0.029 | 0.576±0.066 | 0.640±0.052 |
| | 0.20 | 16.120±1.751 | 0.520±0.029 | 0.376±0.066 | 0.659±0.052 |
| Gaussian Blur | 1.00 | 25.159±1.999 | 0.744±0.059 | 0.929±0.017 | 0.320±0.048 |
| | 2.00 | 22.365±1.875 | 0.646±0.073 | 0.849±0.033 | 0.434±0.057 |
| | 4.00 | 19.738±1.739 | 0.558±0.080 | 0.715±0.051 | 0.579±0.065 |

Table 15: Running times of our IDA-RD with PULSE and SRFlow as $f_{us}$ (Eq. 2) respectively. $N_X$: the number of images in test dataset $X$ in Eq. 2.

| $N_X$ | PULSE | SRFlow |
|---|---|---|
| 300 | 3h08min | 18min |
| 600 | 6h10min | 35min |
| 900 | 9h08min | 55min |

Table 16: Additional results of NIQE and BRISQUE at higher resolutions (lower is better).

| Resolution | LR | $\sigma = 1.0$ | $\sigma = 2.0$ | $\sigma = 4.0$ |
|---|---|---|---|---|
| 1024×1024 | 3.700 | 4.158 | 5.173 | 6.471 |
| 512×512 | 2.406 | 3.959 | 5.574 | 6.299 |
| 256×256 | 3.047 | 4.611 | 7.133 | 6.792 |
| 128×128 | 18.873 | 18.872 | 18.870 | 18.869 |
| 64×64 | 18.872 | 18.872 | 18.870 | 18.869 |
| 32×32 | 18.873 | 18.869 | 18.870 | 18.867 |

(a) NIQE scores

| Resolution | LR | $\sigma = 1.0$ | $\sigma = 2.0$ | $\sigma = 4.0$ |
|---|---|---|---|---|
| 1024×1024 | 26.792 | 32.827 | 48.971 | 59.043 |
| 512×512 | 19.536 | 33.391 | 57.447 | 63.144 |
| 256×256 | 28.582 | 39.282 | 55.747 | 65.990 |
| 128×128 | 16.045 | 34.423 | 47.017 | 55.166 |
| 64×64 | 41.360 | 42.417 | 43.346 | 54.344 |
| 32×32 | 43.458 | 43.458 | 44.015 | 43.668 |

(b) BRISQUE scores

evaluate downscaling algorithms whose output images are of higher quality than those generated by the SR model (*i.e.*, no distortion).

**Future work.** Our framework still requires a ground truth HR image. However, we believe the distortion can be calculated without such a ground truth image. To further validate our IDA-RD measure, in the future we will we use the *meta-measure* methodology (Pont-Tuset & Marques, 2013; Fan et al., 2019), in which secondary, easily quantifiable measures are constructed to quantify the performance of a less easily quantifiable measure.

## A.7 ABLATION STUDY OF $N_X$ FOR IDA-RD IMPLEMENTED WITH PULSE

As Table 17 shows, we also investigate how many images are required in the test dataset $X$ consisting of high-resolution images to achieve a robust estimation of IDA-RD implemented with PULSE (Menon et al., 2020). Similar to those in the main paper, it can be observed that the results become stable when $N_X \geq 900$, which further justifies our choice of $N_X = 900$ for IDA-RD.

Table 17: Ablation study of $N_X$ for IDA-RD implemented with PULSE. Synthetic image downscaling methods with Contrast Decrease with $\sigma = 0.75$ (DG1); Gaussian Noise with $\sigma = 0.05$ (DG2); mixed noise consisting of Gaussian Blur with $\sigma = 1.0$, Contrast Decrease with $\sigma = 0.75$, and Gaussian Noise with $\sigma = 0.05$ (DG3); are used in the experiments.

| $N_X$ | 30 | 300 | 600 | 900 | 1200 | 1500 |
|---|---|---|---|---|---|---|
| DG1 | 0.351±0.014 | 0.342±0.019 | 0.343±0.012 | 0.339±0.022 | 0.339±0.021 | 0.339±0.023 |
| DG2 | 0.361±0.011 | 0.383±0.011 | 0.374±0.012 | 0.351±0.023 | 0.353±0.022 | 0.352±0.021 |
| DG3 | 0.471±0.011 | 0.483±0.012 | 0.391±0.013 | 0.293±0.019 | 0.289±0.022 | 0.291±0.021 |

## A.8 EXPERIMENTAL RESULTS ON REAL-WORLD DATASETS

Table 18 shows our IDA-RD scores on two real-world datasets: DIV2K (Agustsson & Timofte, 2017) and Flickr30k (Young et al., 2014). It can be observed that our conclusions still hold (N.N. > Perceptual > L0-regularized > DPID > Bicubic > Bilinear), which further justifies the validity of the proposed IDA-RD measure. Note that both experiments are conducted with a scaling factor of $4\times$ as we observed SRFlow become unstable for higher scaling factors (Fig. 2). For stable uses of SRFlow, we intentionally used domain-specific datasets in the main paper. Note that all state-of-the-art image downscaling methods (*i.e.*, Perceptual, L0-regularized, DPID) used in our experiments are general ones that are applicable to all domains (*i.e.*, not tuned for specific domains).

Table 18: IDA-RD scores on two real-world datasets: DIV2K (Agustsson & Timofte, 2017) and Flickr30k (Young et al., 2014). UD: "unknown downscaled" images provided by DIV2K. A scaling factor of $4\times$ is used for both datasets as we observed higher scaling factors makes SRFlow unstable.

| | Bicubic | Bilinear | N.N. | DPID | Perceptual | $L0$-reg. | UD |
|---|---|---|---|---|---|---|---|
| DIV2K | 0.157±0.073 | 0.129±0.089 | 0.374±0.079 | 0.216±0.057 | 0.336±0.068 | 0.226±0.072 | 0.355±0.128 |
| Flickr30K | 0.263±0.102 | 0.239±0.112 | 0.452±0.105 | 0.357±0.097 | 0.367±0.080 | 0.364±0.103 | – |

Figure 2: SRFlow becomes unstable for a scaling factor of $8\times$ on real-world datasets, *e.g.*, DIV2K (Row 1), while such cases never happen for domain-specific datasets, *e.g.*, FFHQ (Row 2).

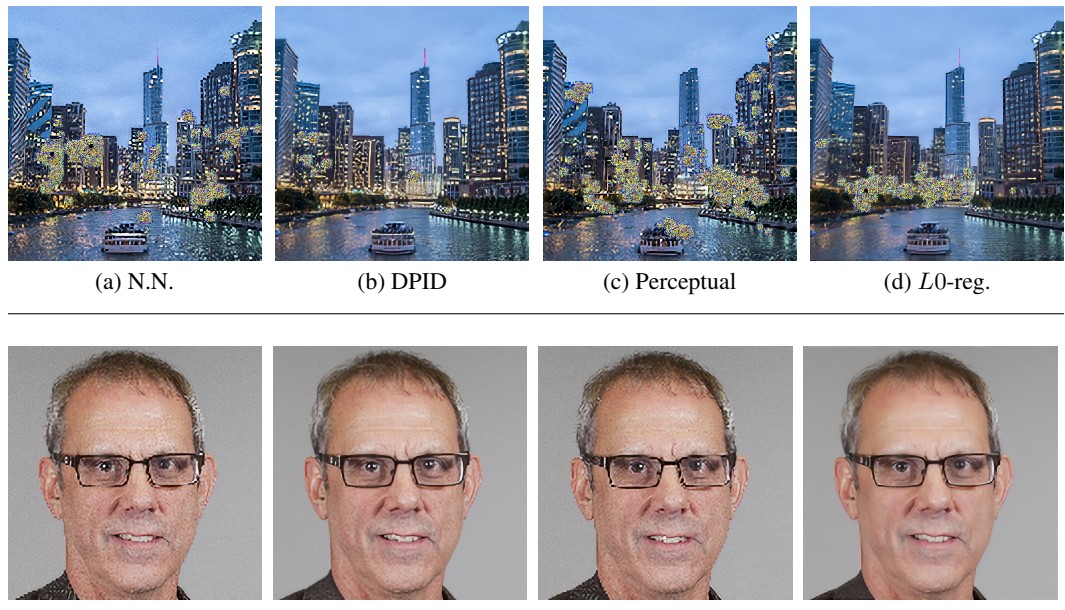

| (a) N.N. | (b) DPID | (c) Perceptual | (d) $L0$-reg. |
|---|---|---|---|

| (e) N.N. | (f) DPID | (g) Perceptual | (h) $L0$-reg. |
|---|---|---|---|

## A.9 ADDITIONAL ABLATION STUDY OF $f_{us}$

As Table 19 shows, we tested our IDA-RD measure with some other choices of state-of-the-art SR methods: BSRGAN (Zhang et al., 2021), RSR (Castillo et al., 2021) and Real-ESRGAN (Wang et al.). However, all these methods are blind but **non-stochastic** (Sec. 4.5), which do not satisfy the requirement of our IDA-RD measure and generate less informative results. Specifically, the results of BSRGAN and Real-ESRGAN are less distinguishable among different downscaling methods; the results of RSR are slightly better but still not comparable to SRFlow.

## A.10 QUALITATIVE EVALUATION OF EXISTING DOWNSCALING METHODS

As Fig. 3 shows, state-of-the-art image downscaling methods achieve better perceptual quality by "exaggerating" perceptually important features in the original image (*e.g.*, building lights, water

Table 19: Additional ablation study of $f_{us}$, the image upscaling algorithms. Following Sec. A.8, we use the DIV2K dataset and a scaling factor of 4×. BSRGAN (Zhang et al., 2021), RSR (Castillo et al., 2021) and Real-ESRGAN (Wang et al.) are blind but non-stochastic SR methods (Sec. 4.5), which do not satisfy the requirement of our IDA-RD measure and generate less informative results.

| | Bicubic | Bilinear | N.N. | DPID | Perceptual | $L0$-reg. |
|---|---|---|---|---|---|---|
| BSRGAN | 0.010±0.008 | 0.011±0.008 | 0.024±0.022 | 0.013±0.011 | 0.025±0.018 | 0.011±0.008 |
| RSR | 0.231±0.071 | 0.208±0.095 | 0.423±0.132 | 0.288±0.099 | 0.379±0.123 | 0.231±0.071 |
| Real-ESRGAN | 0.014±0.010 | 0.015±0.011 | 0.026±0.022 | 0.016±0.012 | 0.026±0.017 | 0.017±0.013 |

reflections), thus leading to over-exaggeration in the upscaled images. As a result, they have lower IDA-RD scores than bicubic and bilinear downscaling.

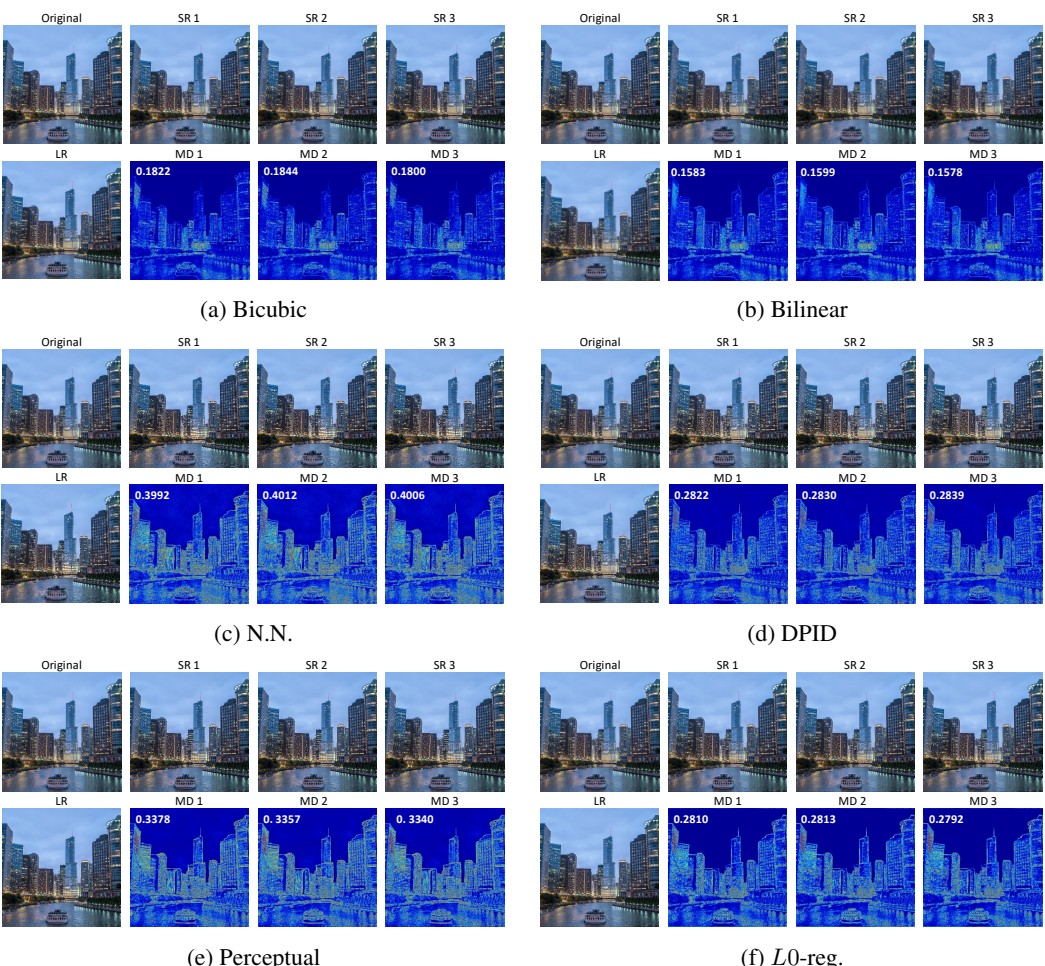

Figure 3: Qualitative evaluation of existing image downscaling methods. Original: the input HR image; LR: the downscaled LR image; SR1, SR2, SR3: three instances of upscaled images; MD1, MD2, MD3: difference map visualizations of (SR1, Original), (SR2, Original), and (SR3, Original), respectively. The white numbers on the left-top corners: the corresponding LPIPS scores of the difference map visualizations. State-of-the-art image downscaling methods (DPID, Perceptual and $L0$-reg.) achieve better perceptual quality by "exaggerating" perceptually important features in the original image (*e.g.*, building lights, water reflections), thus leading to over-exaggeration in the upscaled images and lower IDA-RD scores.

