# OpenReview forum: "Deep Generative Model based Rate-Distortion for Image Downscaling Assessment"
_ICLR.cc/2023/Conference — Submitted to ICLR 2023_

### Official Review · Reviewer_CnsW · 2022-10-24

**Confidence:** 2
**Correctness:** 3
**Technical Novelty And Significance:** 3
**Empirical Novelty And Significance:** 3
**Recommendation:** 6

**Clarity, Quality, Novelty And Reproducibility:**

The work contributes some novel ideas, such as the notion of using "information loss during downscaling via SR networks as a quantitative metric." Some of the descriptions are missing with respect to the experiments (for example, under "Invalidity of Non-blind and Non-stochastic SR method," it is not clear if the author used pre-trained networks for ESRGAN (Wang et al., 2018) and SR3 (Saharia et al., 2022))

**Strength And Weaknesses:**

Strengths

1. The paper addresses an interesting problem of measuring the downscaling methods
2. The paper does contribute some new ideas to tackle the problem at hand
3. The proposed methods outperform some of the other existing metrics on domain-specific image datasets

Weaknesses

1. Missing comparisons: There are some other upscaling super-resolution (SR) functions that are likely to handle the unseen degradations well, for example, methods in [p1-p3]. The experiments should be conducted with these methods as well.

[p1] Designing a Practical Degradation Model for Deep Blind Image Super-Resolution, ICCV 2021
[p2] Real-esrgan: Training real-world blind super-resolution with pure synthetic data, ICCVW 2021
[p3] Generalized Real-World Super-Resolution Through Adversarial Robustness, ICCVW 2021


2. Conclusions from little pieces of evidence: (Refer: Section 4.3) Based on the results, authors argue that SOTA downscaling methods (Weber et al., 2016; Liu et al., 2017; Oeztireli & Gross, 2015) actually lose more information than Bilinear downscaling. However, this conclusion comes from the assumption that the proposed method is optimal for measuring the distortions of any arbitrary downscaling operation. This assumption is unlikely to be true and needs further evaluation, analysis, and comparison before coming to a conclusion. To this end, I would recommend adding “comparison: with mean opinion score (i..e, feedback from humans based on perceptual understanding)”, “analysis: such as picking some examples that best reveal the distortion effects and then arguing why the SOTA methods perform poorly in comparison to bilinear”, and  “evaluation: ensure fair evaluation since the SOTA methods might have fine-tuned for say natural images or specific types of images or distortions”. Altogether we need much more experimental evidence in support of this kind of argument that actually contradicts the common beliefs in the literature. Moreover, the experiments are done only with domain-specific images, which also makes adds to the fact that the proposed method is yet an “incomplete metric”.

3. Clarity on the choices of methods:
- Regarding the GAN generator-based method: Based on Eq. 5, the proposed method employs the downscaling method that we wish to evaluate to generate the LR image for training the super-resolution (SR) method. Such training can be done with any fully-supervised SR method. Henceforth, it is not clear why other SR methods would fail to do the same job or what the limitations are in using other methods. There should also be experiments to support such arguments/failure cases. There seem to be some comparisons under “Invalidity of Non-blind and Non-stochastic SR method” it is not clear if the experiments were done with pre-trained networks or the ones trained with “f_ds” as in Eq. 5.
- Regarding the SRflow method: This method employs bicubic downscaling for training the SR network is more attractive since we can avoid retraining the SR network to evaluate any new downscaling method. However, as I have mentioned before, a more “generalizable to unseen degradations SR network” could be [p1-p3] and should therefore be included in experiments and comparisons.

4. Why only domain-specific datasets in experiments and comparisons?

The authors seem to have used only the face images for all the comparisons. Why are the natural images not included in the study? Or is it like the authors were trying to tackle metrics for certain domain-specific images alone?

**Summary Of The Paper:**

This paper proposes a new metric, namely IDA-RD, to evaluate image downscaling algorithms quantitatively. To this end, the authors employ the idea that a downscaling algorithm that preserves more details in the resulting low-resolution images should lead to less distorted high-resolution images constructed by state-of-the-art super-resolution methods. Experimental comparisons are made over various downscaling operations with different levels of distortions to demonstrate the effectiveness of the proposed method.

**Summary Of The Review:**

While the work presents some interesting ideas, many comparisons and experiments are either missing or are not done fairly. The overall evaluation of the method seems to lack multiple aspects, which makes the paper below the acceptance threshold.

Comments after Author Response:
Based on the author's response I am changing my rating from Reject to Accept.

---

> ### Author Response · Authors · 2022-11-18
> **Reply to Reviewer CnsW**
>
> We thank the reviewer for their comments and hope our response addresses the concerns raised.
>
> **Q1.** Missing comparisons.
>
> Thanks for the suggestion and we have included experiments with these methods in **Table 19, Appendix A.9** of the revised paper, which further demonstrates the versatility of the proposed measure across different choices of image upscaling functions $f_{us}$.
>
> **Q2.** Conclusions in Sec. 4.3 contradict the common belief in literature and requires more evidence.
>
> First, we would like to clarify a misunderstanding: quoting our responses to Reviewer o28g (Q2) and Reviewer mBvw (Q5), “we would like to clarify that the aim of the proposed measure is **not** to approximate human perception of downscaled images but rather the extent to which they preserve the information of their corresponding HR images, which is objective and orthogonal to perceptual quality. Thus, our results do not contradict previous ones of state-of-the-art methods but provides a new dimension to evaluate image downscaling results. We believe that a good image downscaling method should be of high perceptual quality, while retaining as much information as possible from the original HR image (our measure).” In short, our measure is orthogonal to human perception and does not contradict previous literature. Thus, we believe it is not useful and unnecessary to repeat the same user studies in previous literature, e.g., the “comparison: with mean opinion score (i.e., feedback from humans based on perceptual understanding)” as suggested.
>
> For the “analysis: such as picking some examples that best reveal the distortion effects and then arguing why the SOTA methods perform poorly in comparison to bilinear”, we have included the analysis in **Fig. 3, Appendix A.10** of the revised paper.
>
> For the “evaluation: ensure fair evaluation since the SOTA methods might have fine-tuned for say natural images or specific types of images or distortions”, we believe this is not necessary as all the SOTA methods used (i.e., DPID, Perceptual and L0-reg.) are general ones that are applicable to all domains (i.e., not tuned for specific domains). This indicates that our experiments are completely fair, even on domain-specific datasets.
>
> In addition to domain-specific datasets, we have also included additional experiments on DIV2K and Flickr30K in **Table 18, Appendix A.8** of the revised paper. The results support our major claims as well.
>
> **Q3.** Clarity on the choices of methods: (i) Regarding the GAN generator-based method: it is not clear why other fully-supervised SR methods would fail to do the same job or what the limitations are in using other methods. The comparisons under “Invalidity of Non-blind and Non-stochastic SR method” is not clear if the experiments were done with pre-trained networks or the ones trained with “f_ds” as in Eq. 5. (ii) Regarding the SRFlow method: as mentioned before, a more “generalizable to unseen degradations SR network” could be [p1-p3] and should therefore be included in experiments and comparisons.
>
> (i) First, we would like to clarify that **all** image upscaling models used in our work are pretrained ones, including the StyleGAN generator in Eq. (5) and those under “Invalidity of Non-blind and Non-stochastic SR method”. The reason why other fully-supervised SR methods do not work is that: image upscaling is inherently ill-posed that a LR image can generate a distribution of valid HR images [1]. The size of such a HR distribution (i.e., $\hat{x}|x$) measures the information loss during downscaling. However, many fully-supervised SR methods can only generate a single output but not a distribution of valid HR images, thus cannot be used in our measure.
>
> (ii) Thanks for the suggestion and we have included the experiments on the methods mentioned in **Table 19, Appendix A.9** of the revised paper.
>
> [1] Lugmayr, A., Danelljan, M., Gool, L.V. and Timofte, R., 2020, August. SRFlow: Learning the Super-resolution Space with Normalizing Flow. ECCV 2020.
>
> **Q4.** Why only domain-specific datasets in experiments and comparisons?
>
> We used them i) for a fair comparison with the implementation with PULSE/StyleGAN in Ablation Study (Sec. 4.4); ii) for the stable use of SRFlow, as we observed that SRFlow become unstable for higher scaling factors (e.g., 8x) on real-world datasets. However, this has never happened for domain specific datasets. Please see **Fig. 2, Appendix A.8** of the revised paper for more details.
>
> We have also included additional experiments on DIV2K and Flickr30K in **Table 18, Appendix A.8** of the revised paper. The results support our major claims as well.

---

> > ### Comment · Reviewer_CnsW · 2022-12-03
> > **Final Comments**
> >
> > I have reviewed the author's response and the revised version of the paper. All of my concerns have been addressed adequately. I would upgrade my rating to Accept. Thanks to the authors for all the clarifications.

---

> > > ### Author Response · Authors · 2022-12-03
> > > **Many thanks**
> > >
> > > Thank you so much for upgrading your rating to Accept!

---

### Official Review · Reviewer_mBvw · 2022-10-24

**Confidence:** 4
**Correctness:** 2
**Technical Novelty And Significance:** 2
**Empirical Novelty And Significance:** 2
**Recommendation:** 3

**Clarity, Quality, Novelty And Reproducibility:**

This paper is very sloppy in its mathematical presentation. After many readings, I can probably understand the author's intention. But it's not simple. I need to guess what the author actually intended.

**Strength And Weaknesses:**

Weaknesses:
1. The statement of Eq (2) is very confusing. $Q$ denotes conditional distribution. The distribution of what? Of the lossy image reconstruction function $f$? Is the $Q$ can be written by $Q(f|f_{ds})$? Or should it be $Q(x’|x)$ according to the $P(z)\to Q(x’|x)$. I don’t know what the random variable is and I don’t know why there is a distribution. From any aspect, Eq (2) seems to make no sense. Moreover, is $x’$ follows a distribution? How can D measure between $x$ and $x’$? Is D a metric of distributions?
2. The many-to-many part is also very confusing. I know that the generative model generates a distribution or something, but it is not one-to-many. For everything that is generated by the model, we need a latent code. With the latent code, it is not one-to-many. And I don’t think we need a many-to-many statement here.
3. The above two questions make Eq (4) even more confusing.
4. In Eq (5), what is $z*$? And the mathematical is not presented in the right way.
5. For the experiments, There is no conclusive evidence proving that the proposed method is consistent with the human perception results. It is not clear whether the proposed method is linear in evaluation, or whether it is order-preserving. Experiments on synthetic images can provide limited evidence. However, due to the very large degradation of the synthetic images, it is difficult to make such a big difference when evaluating the downsampling method. So these experiments are not enough to provide evidence that their methods are consistent with human perception results.
6. Are there any other applications of the proposed method? For example as a clue to know new downsampling methods?

**Summary Of The Paper:**

This paper proposes a new method to measure image downscaling algorithms. The core idea is to upsample the downsampled image using blind/one-to-many generative-model-based super-resolution methods and compare the upsampled image with the original image. The similarity will be used as the final result. The authors show results on some synthetic data and compare different methods.

**Summary Of The Review:**

The main problem with the paper is the writing, and imperfect experimental proof of the performance and significance.

---

> ### Author Response · Authors · 2022-11-18
> **Reply to Reviewer mBvw**
>
> We thank the reviewer for their comments and hope our response addresses the concerns raised.
>
> We have rephrased Sec. 3 in the revised paper to make it clearer. Please see our reply to the specific points below.
>
> **Q1.** Eq. (2). (i) Meaning of $Q$. Eq. (2) seems to make no sense. (ii) Does $\hat{x}$ follow a distribution? (iii) How can D measure between $x$ and $\hat{x}$? Is D a metric of distributions?
>
> (i) $Q_f(\hat{x}|x)$ or $Q$ for short is the probability density function (PDF) of reconstructed HR images $\hat{x}$ conditioned on an input HR image $x$ with respect to a given lossy image reconstruction function $f$ that $\hat{x} = f(x) = f_{us}(f_{ds}(x))$, where $f_{us}$ and $f_{ds}$ denote image upscaling and downscaling functions respectively. We have removed Eq. (2) and rephrased other equations in Sec. 3 accordingly (without using $Q$) in the revised paper.
>
> Also, there are no random variables in these equations as Sec. 3.1 only shows the concepts rather than the specific implementations. The random variables (i.e., latent codes $\mathbf{z}$) are introduced in the implementation of $f_{us}$ with deep generative models in Sec. 3.2 of the main paper.
>
> (ii) Yes, $\hat{x}$ follows the conditional distribution $\hat{x}|I_{LR}$ or $\hat{x}|f_{ds}(x)$, and hence $\hat{x}|x$, where $I_{LR}=f_{ds}(x)$ is LR image downscaled by $f_{ds}$, $x$ is the input HR image. This distribution comes from the information loss during image downscaling, which results in a distribution of valid reconstructed HR images.
>
> (iii) $D_Q$ is an abstract metric between two distributions/image sets. $D$ is its implementation, a metric between two images of the same resolution (e.g., PSNR, LPIPS). We have clarified this in Sec. 3 of the revised paper.
>
> **Q2.** The many-to-many part is very confusing. Generative models are not one-to-many as they generate one image per latent code.
>
> We believe there is an important misunderstanding: our “one-to-many”, “many-to-many” is for the image upscaling function $f_{us}$ but not generative models. Specifically, by “one-to-many” we meant the fact that image upscaling is inherently ill-posed (“one” input LR image may correspond to “many” HR images); by “many-to-many” we meant that the input LR images of $f_{us}$ could be downscaled with a variety of methods (e.g., bicubic, nearest neighbour), thus changing the problem from “one-to-many” to “many-to-many” that is more challenging. Please note that the inputs to $f_{us}$ are LR images but not latent codes. The latent codes are only used to estimate the conditional distribution $\hat{x}|x$ given an input LR image $x$.
>
> In short, the “many-to-many” means that the input LR images of $f_{us}$ might be from “many” image downscaling methods, and there could be “many” valid output HR images due to the inherent ill-posed nature of image upscaling.
>
> **Q3.** The above two questions make Eq (4) even more confusing.
>
> We hope our revised Sec.3 and responses to the two questions above help the understanding of Eq. (4) (now Eq. (3) in the revised paper).
>
> **Q4.** Eq. (5). What is $z^*$? Clarification on the mathematics.
>
> As a common practice in optimization, we use $z^*$ to denote the optimized latent code. We have also rephrased the math to make it clearer in Sec. 3.
>
> **Q5.** There is no conclusive evidence proving that the proposed method is consistent with the human perception results.
>
> We believe this is a key misunderstanding. Quoting our response to Reviewer o28g (Q2), “we would like to clarify that the aim of the proposed measure is **not** to approximate human perception of downscaled images but rather the extent to which they preserve the information of their corresponding HR images, which is objective and orthogonal to perceptual quality. Thus, our results do not contradict previous ones of state-of-the-art methods but provide a new dimension to evaluate image downscaling results. We believe that a good image downscaling method should be of high perceptual quality, while retaining as much information as possible from the original HR image (our measure).” Thus, our method is orthogonal to human perception and does not need to be consistent with it.
>
> **Q6.** Are there any other applications of the proposed method? For example as a clue to know new downsampling methods?
>
> Yes, our measure can work as a clue to know new image downscaling methods through the lens of information loss, as suggested. Our method may also help devise better semantic translation methods for the visually impaired [1].
>
> [1] Xia, X., He, X., Feng, L., Pan, X., Li, N., Zhang, J., Pang, X., Yu, F. and Ding, N., 2022. Semantic Translation of Face Image with Limited Pixels for Simulated Prosthetic Vision. Information Sciences, 609, pp.507-532.

---

> ### Comment · Reviewer_mBvw · 2022-11-19
> **Follow-up Discussion**
>
> I have read the revised paper and the author's response. The description of the new version is much easier to understand than the old version. Of course, this may be because I am also thinking about the content of this paper these days. The symbols look more acceptable.
>
> I think now is a good time to discuss some new issues. I realize that the window for authors to revise their manuscripts has passed, but this will not affect my judgment, and I would not ask to see a revised version if the authors could clarify something in the discussion.
>
> 1. I agree with the author's core idea. Better downsampling methods should preserve more information. But its main technical focus is blind super-resolution and generative models. I am very familiar with both technologies. A general super-resolution model utilizes a specifically downsampled model (e.g., bicubic) and overfits the low-resolution image produced by this downsampling. At this time, any tiny information in the image belongs to the category of super-resolution model utilization. It is understandable that such a property can be used to detect the information loss of the downsampled image. But the blind super-resolution models, to achieve the effect of "equal treatment" of low-resolution images generated by different downsampling, they need to ignore specific patterns related to downsampling. In short, the blind super-resolution model does not utilize the information in all low-resolution images but selectively utilizes them. At this time, this article's core idea has been questioned: What if the difference between different low-resolution images is ignored by the blind super-resolution model? Experiments have not allayed this concern of mine.
> 2. I also have concerns about the discussion of generative models. I don't think generative models are even necessary. Assume an SR generative model and an SR model trained with l2 loss. To a certain extent, although not rigorous, I can think that the result of the l2 model is the average of the multiple results of the generative model. The SR method based on the generative model, like the blind method, will ignore the details of many low-resolution images. Their results are not guaranteed to be faithfully reconstructed. I don't know how to solve this problem.
> 3. If there is a suitable experiment to prove the method in the article, even if it is not rigorous, it can also be used as a reference to a certain extent. But there is no such label for ground truth information retention. The degradation of the validation experiments in this paper is quite large, which cannot directly explain the effectiveness of this method when evaluating different downsampling.
>
> I am open to further discussion.
>
> Reviewer mBvw,

---

> > ### Author Response · Authors · 2022-11-23
> > **Reply to "Follow-up Discussion"**
> >
> > We thank the reviewer very much for their appreciation of our response and the revised paper, their time in understanding our paper these days, and their willingness for further discussion!
> >
> > **Q1.** The blind super-resolution model does not utilize the information in all low-resolution images but selectively utilizes them. What if the difference between different low-resolution images is ignored by the blind super-resolution model? Experiments have not allayed this concern of mine.
> >
> > Thanks for the appreciation of our core idea and the insightful comments. Since "experiments have not allayed this concern of mine", we will explain the conceptual rationale of our idea below.
> >
> > First, we hope to clarify that our IDA-RD measure requires more than a blind super-resolution method but a **blind** and **stochastic** (Sec. 3.1 of the main/revised paper, second bullet point) super-resolution method, which alleviates the effect of "equal treatment" that "ignore specific patterns related to downsampling". This is one of our key insights.
> >
> > Specifically, the **stochasticity** requirement relaxes the output of blind super-resolution from a single ground truth HR image to a distribution of many valid HR images. This alleviates the "equal treatment" effect as different downscaling algorithms have **different** such distributions rather than the same ground truth image (i.e., the outputs of different input downscaled images are no longer "equal"). Thus, conceptually, blind and stochastic SR algorithms should not ignore the differences between different LR images.
> >
> > **Q2.** I don't think generative models are even necessary. I can think that the result of the l2 model is the average of the multiple results of the generative model. The SR method based on the generative model, like the blind method, will ignore the details of many low-resolution images.
> >
> > Thanks for the comments. We use generative model based SR algorithms as to our knowledge, they are the only ones that satisfy the **stochasticity** requirement discussed in our above response to Q1.
> >
> > In addition, as discussed in our response to Q1, since the **stochastic** generative model based SR methods relax the output of blind super-resolution from a single ground truth HR image to a distribution of many valid HR images, they alleviate the "equal treatment" effect and will not ignore the details of LR images.
> >
> > **Q3.** (i) No ground truth to compare. (ii) Large degradation cannot directly explain the effectiveness of this method when evaluating different downsampling.
> >
> > (i) Yes, we agree that image downscaling assessment is very challenging as there are no ground truth LR images to compare. However, we hope to clarify that this is the motivation of this work and the rationale for our IDA-RD measure to include a proxy task of lossy image compression (following the rate-distortion theory) via image downscaling and upscaling. Through this proxy task, we move the comparison of ground truth from between LR images to between the reconstructed and the input HR images. In other words, our IDA-RD uses the input HR image as the ground truth for comparison, which we believe is suitable for evaluation.
> >
> > (ii) We have included more results on smaller degradations below (following Table 1ab in the main/revised paper), which further demonstrate the effectiveness of our method in distinguishing between different levels of small degradations:
> >
> > |Gauss. Blur                          |  Gauss. Noise                    | Contrast Inc.                     | Contrast dec.                   | Quantization               |
> > |  ----                                    |  ----                                   | ----                                   | ----                                  | ----                              |
> > |  (0.01) 0.118$\pm$0.042   | (0.001) 0.118$\pm$0.054 | (1.05) 0.120$\pm$0.0324 | (0.95) 0.113$\pm$0.032  | (19) 0.111$\pm$0.035 |
> > |  (0.25) 0.202$\pm$0.044   | (0.005) 0.291$\pm$0.062 | (1.15) 0.121$\pm$0.0296 | (0.90) 0.119$\pm$0.032  | (18) 0.182$\pm$0.038 |
> > |  (0.50) 0.267$\pm$0.050   | (0.010) 0.318$\pm$0.061 | (1.20) 0.130$\pm$0.0292 | (0.80) 0.131$\pm$0.032  | (17) 0.193$\pm$0.041 |
> >
> > The numbers in the parentheses denote the degradation parameters.

---

> > > ### Comment · Reviewer_mBvw · 2022-11-26
> > > **Reply to "Reply to "Follow-up Discussion""**
> > >
> > > Thanks for the author's reply. I still can't fully accept the author's explanation, e.g., "This alleviates the "equal treatment" effect as different downscaling algorithms have different such distributions rather than the same ground truth image".
> > >
> > > I think that blind SR may have certain flaws, but the author thinks these flaws will not have a lot of impacts. It is ineffective to discuss concepts in isolation from facts. This paper presents a method for evaluating a class of algorithms. Its evaluation must be adequately verified to be reliable. I don't think there is a good reason for the author's argument to be reliable, because it is very difficult to provide references for the experiments in this paper. The authors provide some experiments on "small" degradations. But how do we prove that these experiments are sufficient to model the difference made by the downsampling algorithms? I am considering three new types of experiments: (1) Degradation before downsampling will make more sense. (2) Investigate the minimum degradation that causes differences in IDA-RD values, and evaluate the stability of IDA-RD before producing statistically significant differences. (3) Conduct some visualization studies to show that it is indeed a flaw in the downsampling algorithm that is causing the numerical difference.
> > >
> > > As for my second question. I understand the stochasticity thing. But Eq2 in the paper shows that expectation is actually more important. If I can get this expectation, stochasticity is not necessary.

---

> > > > ### Author Response · Authors · 2022-12-01
> > > > **Reply to "Reply to "Reply to "Follow-up Discussion"""**
> > > >
> > > > We thank the reviewer very much for their additional questions. As requested, we have conducted the following three experiments which we hope will address their concerns.
> > > >
> > > > **Experiment 1.** Degradation before downsampling will make more sense.
> > > >
> > > > | Gauss. Blur          | Gauss. Noise          | Contrast Inc.        | Contrast dec.          | Quantization         |
> > > > |  ----                |  ----                 | ----                 | ----                   | ----                 |
> > > > |(1.0) 0.321$\pm$0.048 |(0.05) 0.480$\pm$0.031 |(1.5) 0.234$\pm$0.042 |(0.75) 0.330$\pm$0.047  | (15) 0.162$\pm$0.015 |
> > > > |(2.0) 0.432$\pm$0.050 |(0.10) 0.641$\pm$0.052 |(2.0) 0.317$\pm$0.043 |(0.50) 0.644$\pm$0.070  | (10) 0.205$\pm$0.013 |
> > > > |(3.0) 0.579$\pm$0.055 |(0.20) 0.658$\pm$0.052 |(2.5) 0.462$\pm$0.043 |(0.25) 0.669$\pm$0.034  | (5 ) 0.464$\pm$0.054 |
> > > >
> > > > It can be observed that applying degradation before downscaling yields similar results to applying degradation after downscaling. We therefore conclude that either approach yields valid synthetic downscaling methods. We will incorporate this into our revised paper to make the evaluation more comprehensive. Thanks again for this suggestion.
> > > >
> > > > **Experiment 2.** Investigate the minimum degradation that causes differences in IDA-RD values, and evaluate the stability of IDA-RD before producing statistically significant differences.
> > > >
> > > > | Gauss. Blur             | Gauss. Noise            | Contrast Inc.          | Contrast dec.           | Quantization         |
> > > > |  ----                   |  ----                   | ----                   | ----                    | ----                 |
> > > > |(0.0001) 0.111$\pm$0.034 |(0.0001) 0.110$\pm$0.029 |(1.001) 0.111$\pm$0.034 |(0.999) 0.111$\pm$0.034  | (19) 0.111$\pm$0.035 |
> > > > |(0.0005) 0.112$\pm$0.034 |(0.0005) 0.110$\pm$0.029 |(1.005) 0.111$\pm$0.034 |(0.995) 0.111$\pm$0.034  | (18) 0.182$\pm$0.038 |
> > > > |(0.0010) 0.113$\pm$0.035 |(0.0010) 0.118$\pm$0.054 |(1.010) 0.115$\pm$0.029 |(0.990) 0.112$\pm$0.031  | (17) 0.193$\pm$0.041 |
> > > > |(0.0050) 0.113$\pm$0.035 |(0.0030) 0.118$\pm$0.062 |(1.050) 0.120$\pm$0.032 |(0.950) 0.113$\pm$0.032  | ------------         |
> > > > |(0.0100) 0.113$\pm$0.036 |(0.0040) 0.203$\pm$0.062 |(1.100) 0.126$\pm$0.029 |(0.900) 0.119$\pm$0.032  | ------------         |
> > > > |(0.0500) 0.114$\pm$0.034 |(0.0050) 0.291$\pm$0.062 |(1.150) 0.126$\pm$0.029 |(0.850) 0.123$\pm$0.031  | ------------         |
> > > > |(0.1000) 0.118$\pm$0.042 |(0.0100) 0.318$\pm$0.061 |(1.200) 0.130$\pm$0.029 |(0.800) 0.131$\pm$0.032  | ------------         |
> > > > |(0.2500) 0.202$\pm$0.043 | ------------            | ------------           |------------             | ------------         |
> > > > |(0.3000) 0.214$\pm$0.044 | ------------            | ------------           |------------             | ------------         |
> > > >
> > > > The table above shows the minimum degradations that cause differences in IDA-RD values (e.g., for Gauss. Blur, when the degradation parameter changes from 0.0001 to 0.0005, the IDA-RD slightly increases from 0.111$\pm$0.034 to 0.112$\pm$0.034), indicating that our IDA-RD is stable against small degradations. Please note that the baseline IDA-RD, i.e., no degradation, is 0.110.
> > > >
> > > > **Experiment 3.** Conduct some visualization studies to show that it is indeed a flaw in the downsampling algorithm that is causing the numerical difference.
> > > >
> > > > We anonymously share examples of images downscaled by different methods below:
> > > > https://www.dropbox.com/s/mjqe1unkv1jcone/Example_images.pdf?dl=0
> > > >
> > > > It can be observed that state-of-the-art (SOTA) image downscaling methods improve the perceptual quality by selectively "enhancing" image features (DPID explicitly mentioned that it "assigns larger weights to pixels that deviate more from their local image neighborhood" [1]), e.g., the glasses frames and clothes patterns in Fig. (i-c,d,e,f); the tessellation gaps in Fig. (ii-c,d,e,f); the hair and watermelon seeds (clothes pattern) in Fig. (iii-c,d,e,f). Nevertheless, selectively "enhancing" perceptually-important features means **downweighting all other features**, resulting in higher uncertainty (i.e., information loss) when reconstructing other features during SR. Since the number of perceptually-important features is typically less than the number of other features, SOTA image downscaling methods lose more information, resulting in higher IDA-RD scores. Please note that N. N. shares a similar idea but uses a very simple "selection" method, thus losing a large amount of information as well.
> > > >
> > > > [1] Weber, N., Waechter, M., Amend, S.C., Guthe, S. and Goesele, M., 2016. Rapid, detail-preserving image downscaling. ACM Transactions on Graphics (TOG), 35(6), pp.1-6.

---

> > > > ### Author Response · Authors · 2022-12-01
> > > > **Reply to "Reply to "Reply to "Follow-up Discussion"""  (Cont'd)**
> > > >
> > > > **Q1.** I understand the stochasticity thing. But Eq2 in the paper shows that expectation is actually more important. If I can get this expectation, stochasticity is not necessary.
> > > >
> > > > Many thanks for the acknowledgment of our explanation of stochasticity. However, we hope to clarify that our "stochasticity" describes the nature of the SR methods we used (i.e., the SR method should be blind and stochastic) but not the computation of our IDA-RD measure.
> > > >
> > > > In terms of computation, we agree that it will be much more computationally-efficient if there are some methods to get the expectation directly, like the kernel trick for inner product computation in high dimensional space. We believe this will be interesting future work for our IDA-RD measure.

---

### Official Review · Reviewer_SxNM · 2022-10-26

**Confidence:** 3
**Correctness:** 3
**Technical Novelty And Significance:** 3
**Empirical Novelty And Significance:** 3
**Recommendation:** 6

**Clarity, Quality, Novelty And Reproducibility:**

Clarity: Good to follow and understand.
Quality: High quality writing and presentation.
Novelty: Somehow novel to use rate distoration model.
Reproduce: This work seems to be reproducedable.

**Strength And Weaknesses:**

Strenght:
1. Rate distortion for evluating down-scaler is interesting.
2. Comprehsive analysis of different down-scalers and different kinds of degradations.

Weaknesses
1. Only SRflow and Pulse sr networks are tested.
2. No realworld datasets are tested. only sythesis datasets.

**Summary Of The Paper:**

This paper is interesting. It gives an indepth analysis of down-scaling methods and degradations for gan sr networks. It uses rate distortion for evalutions and gives some interesting results.

**Summary Of The Review:**

This work worths published given its in depth analysis of down-scaling algorthms using rate distortion. Such results can be useful for scalable video coding in the future codec designs.

---

> ### Author Response · Authors · 2022-11-18
> **Reply to Reviewer SxNM**
>
> We thank the reviewer for their comments and hope our response addresses the concerns raised.
>
> **Q1.** Only SRFlow and PULSE SR networks are tested.
>
> We believe this is a misunderstanding. We also tested ESRGAN and SR3 (Table 10, Sec. 4.5 in the main paper) as negative examples to justify that our measure requires specific SR methods that satisfy two requirements (Sec. 3.1 of the main paper): i) Reconstruct the input image $x$ regardless of the input downscaling algorithm $f_{ds}$; ii) Generate a conditional distribution $Q$ of reconstructed images $\hat{x}$.
>
> We also included additional experiments on BSRGAN, RSR and Real-ESRGAN in **Table 19, Appendix A.9** of the revised paper.
>
> **Q2.** No real-world datasets are tested, only synthetic datasets.
>
> Thanks for the suggestion. We have included additional experiments on DIV2K and Flickr30K in **Table 18, Appendix A.8** of the revised paper. The results support our major claims as well.

---

### Official Review · Reviewer_o28g · 2022-10-26

**Confidence:** 3
**Correctness:** 3
**Technical Novelty And Significance:** 2
**Empirical Novelty And Significance:** 2
**Recommendation:** 6

**Clarity, Quality, Novelty And Reproducibility:**

The proposed method has some novelty, however, the authors should address the questions above.

**Strength And Weaknesses:**

The authors use traditional downscaling algorithms such as bicubic, bilinear, and nearest neighbor interpolation, as well as state-of-the-art downscaling algorithms to test the proposed measure. However, the reviewer does not recommend accepting the current version, because of the following concerns:

1. There are plenty of datasets for real-world super-resolution, i.e., the datasets contain the paired LR and HR images. Therefore, the performance of downsampling can be measured on these datasets by calculating the PSNR/LPIPS between the downsampled images and the original LR images. In comparison with this, what is the benefit of the proposed method?

2. The IDA-RD scores for DPID, Perceptual and L0-reg. are not able to indicate anything, since there is no evidence showing that the predicted score of the proposed measure is correct or not on these methods.

3. What is the relationship between the proposed measure and "rate-distortion"?

4. Why use StyleGAN on PULSE, but SRFlow on other datasets? It doesn't make sense to change the network for different datasets.



**Summary Of The Paper:**

This paper proposes a measure, namely Image Downscaling Assessment by Rate-Distortion (IDA-RD), to quantitatively evaluate image downscaling algorithms.

**Summary Of The Review:**

The authors use traditional downscaling algorithms such as bicubic, bilinear, and nearest neighbor interpolation, as well as state-of-the-art downscaling algorithms to test the proposed measure. However, the reviewer does not recommend accepting the current version, because of the concerns in "Weaknesses" above.

----- After responses -----

The authors addressed some of my concerns.

---

> ### Author Response · Authors · 2022-11-18
> **Reply to Reviewer o28g**
>
> We thank the reviewer for their comments and hope our response addresses the concerns raised.
>
> **Q1.** Why not simply compute PSNR/LPIPS between the downscaled images and the original LR images in real-world super-resolution datasets, i.e., those contain paired LR and HR images?
>
> We would like to clarify that the “original” LR images in such paired super-resolution datasets are **not** ground truth and cannot be used to evaluate downscaled images. Specifically, their LR images are also obtained using software downscaling methods (e.g., bicubic downscaling or some “unknown” downscaling operators [1]), none of which capture the requirements of high-quality downscaling and thus cannot be used as ground truth. Since there is no ground truth, image downscaling assessment has long been a challenging task with little progress.
>
> In this work, we address this challenge by drawing on the rate-distortion theory and recent successes in deep generative models, which sidesteps the requirement of ground truth by measuring the information loss during downscaling. To our knowledge, we are the first to propose a quantitative measure for image downscaling assessment.
>
> [1] Agustsson, E. and Timofte, R., 2017. NTIRE 2017 Challenge on Single Image Super-resolution: Dataset and Study. CVPR workshops 2017.
>
> **Q2.** No evidence showing that the predicted score of the proposed measure is correct or not on DPID, Perceptual and L0-reg.
>
> First, we would like to clarify that the proposed measure is theoretically correct as it is a special case of the widely accepted rate-distortion theory in the context of image downscaling. Please see our response to Q3 below for more details. In practice, the effectiveness of our measure is affected by the deep generative model (e.g., SRFlow) and the distortion metric (e.g., LPIPS) used. To justify the effectiveness of our choices, we conducted experiments on synthetic downscaling methods in Sec. 4.2 of the main paper where the correct results are known in advance, and the results show that our measure is correct. We only apply our measure to DPID, Perceptual and L0-reg. to obtain new insights after knowing it is correct.
>
> Second, we would like to clarify that the aim of the proposed measure is **not** to approximate human perception of downscaled images but rather the extent to which they preserve the information of their corresponding HR images, which is objective and orthogonal to perceptual quality. Thus, our results do not contradict previous ones of state-of-the-art methods but provides a new dimension to evaluate image downscaling results. We believe that a good image downscaling method should be of high perceptual quality, while retaining as much information as possible from the original HR image (our measure).
>
> **Q3.** What is the relationship between the proposed measure and “rate-distortion”?
>
> The proposed measure models image downscaling as the encoding process of a lossy compression problem, where the rate-distortion theory is its theoretical foundation. Specifically, we refer to the number of pixels in a downscaled image as “rate” and the differences between upscaled and the input HR image as the “distortion”. In this way, one aim of image downscaling can be viewed as minimizing the “distortion” given a fixed “rate”, which is often referred to as the rate-distortion optimization (RDO) [2]. Please note that the information loss evaluated by our measure is not the only criterion for image downscaling, as perceptual quality should also be taken into account.
>
> [2] Sullivan, G.J. and Wiegand, T., 1998. Rate-distortion Optimization for Video Compression. IEEE Signal Processing Magazine, 15(6), pp.74-90.
>
> **Q4.** Why use StyleGAN on PULSE, but SRFlow on other datasets? It doesn't make sense to change the network for different datasets.
>
> We believe there are several misunderstandings. First, PULSE is not a dataset but an image upscaling method [3]. Second, we used SRFlow for **all** datasets and did **not** change the network for different datasets. The experiments with PULSE/StyleGAN are only for the ablation study (Sec. 4.4 of the main paper) to justify our choice of $f_{us}$.
>
> [3] Menon, S., Damian, A., Hu, S., Ravi, N. and Rudin, C., 2020. PULSE: Self-supervised Photo Upsampling via Latent Space Exploration of Generative Models. CVPR 2020.

---

> > ### Comment · Reviewer_o28g · 2022-11-18
> > **Real-world super-resolution datasets**
> >
> > Thanks for the response.
> >
> > However, for Q1:
> >
> > The real-world super-resolution datasets are not generated software downscaling methods, but both the LR and HR images are captured by cameras. Please see the RealSR dataset at https://github.com/csjcai/RealSR. The version 3 of RealSR has pairs of LR-HR images (https://drive.google.com/file/d/17ZMjo-zwFouxnm_aFM6CUHBwgRrLZqIM/view). Therefore, the performance of downsampling can be measured on these datasets by calculating the PSNR/LPIPS between the downsampled images and the original LR images. In comparison with this, what is the benefit of the proposed method?
> >
> > I think the reviewer SxNM also has a similar question.

---

> > > ### Author Response · Authors · 2022-11-23
> > > **Reply to additional questions on "Real-world super-resolution datasets"**
> > >
> > > We thank the reviewer very much for their response and additional questions.
> > >
> > > **Q1.** The version 3 of RealSR has pairs of LR-HR images, both are captured by cameras. What is the benefit of the proposed method?
> > >
> > > Thanks for the question. We hope to clarify that LR images captured by cameras should **not** be used as ground truth for image downscaling assessment as they also suffer from information loss depending on the specific camera used, changes in the environment, time, etc.
> > >
> > > Thus, "calculating the PSNR/LPIPS between the downsampled images and the original LR images" is valid only with the assumption that the "original" images captured by cameras are of the highest quality (at least better than **all** downscaling algorithms compared). Previously, there are no quantitative evaluation methods to justify whether this assumption is true. In contrast, **our IDA-RD measure fills the gap as it can also be used to evaluate the information loss of camera-captured LR images**:
> > >
> > >
> > > | Dataset            |    **Camera**       |  Bicubic                 | Bilinear                         | N.N.                        | DPID                     | Perceptual            | $L0$-reg.               |
> > > | ----                  |  ----                     |  ----                       | ----                               | ----                         | ----                       |  ----                       | ----                       |
> > > | RealSR v3 (4x) |0.047$\pm$0.125 |  0.116$\pm$0.052 | 0.114$\pm$0.055         | 0.389$\pm$0.102  | 0.224$\pm$0.079 | 0.341$\pm$0.083 | 0.264$\pm$0.075 |
> > >
> > > It can be observed that i) our IDA-RD measure justifies that the **camera** captured LR images in RealSR v3 are of the highest quality; ii) our conclusions still hold on RealSR v3, i.e., N.N. > Perceptual > L0-regularized > DPID > Bicubic > Bilinear > **Camera**.
> > >
> > > In addition, we anticipate that our measure can also be applied to evaluate different "hardware" downscaling methods, i.e., the images captured by different cameras.

---

> > > > ### Comment · Reviewer_o28g · 2022-11-25
> > > >
> > > > I would upgrade my rating.

---

> > > > > ### Author Response · Authors · 2022-11-26
> > > > > **Many thanks**
> > > > >
> > > > > Thank you so much for upgrading your rating!

---

### Author Response · Authors · 2022-11-18
**To All Reviewers:**

We thank all reviewers for their time and efforts in reviewing our paper but hope to clarify a major misunderstanding as follows.

The aim of the proposed measure is **not** to approximate human perception of downscaled images but rather the extent to which they preserve the information of their corresponding HR images, which is objective and orthogonal to perceptual quality. Thus, our results do not contradict previous ones of state-of-the-art methods but provide a new dimension to evaluate image downscaling results. We believe that a good image downscaling method should be of high perceptual quality, while retaining as much information as possible from the original HR image (our measure). This also indicates that our measure is not suitable for a user study.

We have highlighted our changes in **blue** in the revised paper.

Thank you for reading!

The authors

---

### Decision · Program_Chairs · 2023-01-20

**Decision:**

Reject

**Justification For Why Not Higher Score:**

All reviewers were negative about this paper. Those who acknowledged the author responses were not swayed by those responses. It is especially concerning that the authors seem to misunderstand how ground truth (low res, high res) image pairs are used for evaluating downscaling algorithms: given that such ground truth pairs exist, the usefulness of the proposed method is questionable.

**Justification For Why Not Lower Score:**

N/A

**Metareview: Summary, Strengths And Weaknesses:**

This paper proposes a new metric to evaluate image downscaling algorithms, using the lens of rate distortion theory.

Strengths:
- Use of rate distortion theory to evaluate down-scalers is interesting
- Comprehensive evaluation of different down-scalers under different types of image degradations

Weaknesses:
- Unclear what the advantage of this method is over evaluating PSNR on ground-truth (hi res, lo res) paired data. Authors responded saying that in these paired datasets, lo res data is *not* ground truth because it is produced via some process such as bicubic downsampling. The reviewer responded saying actually, no, both lo-ris and hi-res images are captured by cameras, so they *are* ground truth. Author did not follow up to this.
- Evaluation is only on synthetic data; no real-world data. Authors did upload a revised version of the paper with real-world results that also supports their claims.
- Missing comparisons to some other down-scalers (which the authors added in the revision)

All reviewers were negative. A couple responded to author responses and were not swayed. Seems like clear reject.